# A color-related bias in offside judgments in professional soccer: A matter of figure-background contrast?

Peter Wühr[1]*, Daniel Memmert[2]

1 TU Dortmund University, Dortmund, Germany, 2 German Sports University, Cologne, Germany

* peter.wuehr@tu-dortmund.de

## Abstract

We investigated the impact of outfit colors on the frequency of offside judgments in soccer. In a recent laboratory study, observers made more offside judgments against forwards wearing the outfit of Schalke 04 (blue shirts, white shorts) than against forwards wearing the outfit of Borussia Dortmund (yellow shirts, black shorts), when figure-background luminance contrast was higher for the former team. Here, we investigated whether a similar effect is present in real matches of the German Bundesliga. Study 1 revealed a higher offside score for Schalke 04 than for Borussia Dortmund in matches between these clubs. Studies 2–4 showed higher offside scores for teams wearing a blue/white outfit, and lower offside scores for teams wearing a yellow/black outfit, in their matches against all other Bundesliga teams. Together, results suggest that more offside judgments are made against teams of higher salience, possibly induced by differences in figure-background contrast. Notably, this color-related bias occurred in our study even though a Video-Assistant Referee (VAR) supervised the (offside) decisions of the Assistant Referees.

## Introduction

Football, or soccer, is presumably the most popular sport in the world. For example, according to the "World Football Report 2018" [1], 43% of the respondents were either "interested" or "very interested" in soccer, making it the most popular sport in this survey. The basic rules of the game are simple, and make the game attractive and exciting. One important rule of the game is the "offside" rule, as presented as law 11 of the so-called "laws of the game" [2]. According to this rule, a player P from the attacking team A is in an offside position if—at the moment the ball is played (forward) from a team-mate of P—P is closer to the opponents' goal line than the second last player from the defending team B (which may also include the goal-keeper). Importantly, however, being in an offside position is only penalized by the referee if the corresponding player (P, in our example) is "involved in active play", that is, takes advantage from his position by, for example, scoring a goal.

The main referee is supported by two assistant referees (ARs), one on each side of the field, who have the task of reporting offside positions by raising a flag. In order to optimize the detection of offside positions, ARs are advised to align their position with the "offside" line, an

**Data Availability Statement:** The data reported in this article is publicly available at www.kicker.de. The match statistics of a particular match can be accessed by performing the following steps: (1) access the subpage for matchdays (https://www.kicker.de/bundesliga/spieltag), (2) select the

desired season (e.g., 2018/19), (3) select the desired matchday (e.g., matchday 1), (4) select the desired match (e.g., Bayern Munich vs. Hoffenheim, (5) click on the link "Analyse" (analysis), (6) click on the link "Spieldaten" (match data). Here, you will find the match statistics that were used for our analyses.

**Funding:** The authors received no specific funding for this work.

**Competing interests:** The authors have declared that no competing interests exist.

imaginary line that runs in parallel to the goal line and through the position of the second-last defender. If a member of the attacking team crosses this line, at the moment the ball is played forward, and becomes involved in active play, the AR is supposed to raise his or her flag.

Judging offside is a complex task that requires engagement and coordination of several cognitive processes. This task requires a continuous search for two (or more) players, namely, the defender whose position defines the offside line, and forwards that are close to the offside line, and a continuous comparison of their positions with regard to the goal under attack. A potential strategy of ARs for solving this task might be to determine, first, the player that is at the second last position with regard to the goal of the attacked team, and, secondly, to determine whether this player belongs to the attacking team (offside case) or to the defending team (non-offside case). Because the players are continuously moving, an AR will continuously have to search for players that are in the vicinity of the offside line. Searching and identifying the players around the offside line, and comparing their relative positions, requires covert and overt shifts of visual attention (for reviews on eye movements and visual attention, see [3–5]).

Empirical studies of referee performance in high-level competitions revealed that even professional ARs make, on average, between 10 and 20% errors when judging offside positions (e.g., [6,7]). Because errors in offside judgments may have serious consequences, it is important to understand the sources of these errors, and—if possible—develop measures for preventing them. One may distinguish between team-unspecific and team-specific sources of errors in offside judgments. An example for a team-unspecific source of errors in offside judgments are positional misalignments of the AR (e.g., [8,9]). If, for example, the AR is standing behind the offside line, a forward that is actually in an offside position may appear level with the second-last defender. Positional misalignments represent a team-unspecific source of error in offside judgments because these errors should affect all teams similarly, and they should produce both costs and benefits for every team. In contrast, a team-specific source of error in offside judgments would affect only a particular team, and would consistently produce costs (or benefits) for that team. A systematic preference of a particular (assistant) referee for a particular club would represent an example for a team-specific source of error in offside judgments. It seems desirable to uncover and eliminate both team-specific and team-unspecific sources of error in offside judgments.

In a recent study, Wühr, Fasold, and Memmert [10], investigated the possible impact of team preferences on (errors in) offside judgments in laypersons. Therefore, these authors analyzed and compared performance of two groups of participants when judging offside in (static) displays showing offside or non-offside situations in a "match" between Borussia Dortmund and F.C. Schalke 04, two German soccer clubs with a long-lasting rivalry. Hence, one side of players in the displays wore the typical colors of Borussia Dortmund (i.e., yellow shirts, black shorts), whereas the other side of players in the display wore the typical colors of F.C. Schalke 04 (i.e., blue shirts, white shorts). One group of participants were fans of Borussia Dortmund, whereas the other group of participants were fans of F.C. Schalke 04. The authors hypothesized that team preferences might affect offside judgments by, for example, increasing the number of misses for offside positions of the preferred team as compared to offside positions of the non-preferred team.

Results of their Experiment 1 only partially confirmed the predictions of Wühr et al. [10]. On the one hand, supporters of Borussia Dortmund more often missed offside positions of yellow forwards as compared to blue forwards. On the other hand, supporters of Schalke 04 also missed offside position of yellow forwards more often than offside position of blue forwards, albeit the effect was (significantly) weaker for the latter as compared to the former group. To explain this partly unexpected pattern, the authors assumed two variables had affected the offside judgments in Experiment 1. The first variable was team preference. This variable should

have opposite effects on the accuracy of offside judgments for blue vs. yellow forwards in judgments of the two groups. The second variable was assumed a perceptual variable that increases the accuracy of offside judgments for blue forwards, as compared to yellow forwards, similarly in both groups of participants. In fact, in Experiment 1, the luminance contrast between figure and background had been, inadvertently, larger for blue forwards than for yellow forwards. Two further experiments were conducted to isolate the effects of these two variables.

In their Experiment 2, Wühr et al. [10] used displays in which the luminance contrast between figure(s) and background was equated for the two teams, to isolate the effect of team preferences. Results showed more missed offside positions for the preferred team, supporting the predictions, and the interpretation of the results of Experiment 1. In Experiment 3, Wühr et al. used the same displays as in Experiment 1, allowing for the "luminance" effect, and eliminated the effect of team preferences by recruiting participants that neither supported Borussia Dortmund nor Schalke 04. Results showed more missed offside positions for yellow forwards, as compared to blue forwards, supporting the hypothesis that differences in figure-background luminance contrast—or player salience—can affect offside judgments.

The term "contrast" refers to differences in luminance (or brightness) between two, typically adjacent, areas of the visual field (see, [11] for measures of contrast). Luminance contrasts can occur between figure and ground, or between two figures, and therefore may affect both figure-ground segmentation, and visual object recognition (e.g., [12–14]). The larger the luminance difference (contrast) between figure and background, the easier the figure can be separated from the background and, thus, perceived. A luminance contrast can be positive, when the figure has higher luminance than the background, or negative, when the figure has lower luminance than the background. In the present context, we believe that the critical variable is the difference between the figure-background contrast of team 1 and the figure-background contrast of team 2. Other variables being equal, the figure with higher figure-ground contrast will be detected and processed more easily than the figure with lower figure-ground contrast. In our previous study, the blue shirts were easier perceivable than the yellow shirts because the darker blue shirts contrasted more strongly with the relatively bright background than brighter yellow shirts. In other words: differences in figure-background contrast occur whenever (a) the garments of two teams differ in luminance and (b) the luminance of the background is not in between the luminance of the garments.

The purpose of the present study was two-fold. Firstly, we wanted to test whether the effect of shirt color (i.e., more [correct] offside decisions for blue as compared to yellow forwards) would also occur in real matches between the two clubs. Secondly, on a more general level, we wanted to substantiate the notion that differences in player salience, as induced, for example, by different luminance contrasts between players and background, can systematically affect offside judgments in soccer. Differences in luminance contrast would represent a team-specific source of error in offside judgments because football teams typically wear the same shirt (or outfit) color for long periods of times (e.g., seasons, decades).

Evidence from basic research supports the notion that differences in player salience, as induced by different luminance contrast, may affect offside judgments in soccer. As we have argued before, the task of making correct offside judgments requires searching, finding, and comparing players of different teams near the offside line, and studies have shown that differences in figure-background luminance contrast between targets and distractors affect search performance (e.g., [15–17]). The findings suggest that attention prioritizes items with higher figure-background contrast over items with lower contrast. Thus, stimuli of higher contrast have a higher probability of being inspected first than stimuli of lower contrast (e.g., [16,17]). This attentional bias may have contributed to the observation of more offside judgments for players of Schalke 04 (higher contrast) as compared to players of Borussia Dortmund (lower

contrast) in the Wühr et al. [10] study. A reasonable heuristic for detecting offside positions is to search for the player—among those players that are near the offside line—that is closest to the goal under attack. If this player is a forward, the response will be "offside"; if this player is a defender, the response will be "no offside". Because attention is biased towards the higher-contrast player, participants will more often "find" the higher-contrast player (Schalke 04) than the lower-contrast player (Borussia Dortmund), and therefore the bias would increase offside judgments for forwards of Schalke 04, but decrease offside judgments for players of Borussia Dortmund.

Studies of visual-scan patterns of ARs support the plausibility of the heuristic for detecting offside positions described above. The heuristic would predict that ARs should mostly look at the offside line, and at players (forwards, defenders) that are located close to that line. These predictions fit the empirical results. Several studies revealed that ARs always never fixate the player that plays the ball to his forward, while most of the time the ARs fixate a point in the vicinity of the offside line, or the second-last defender that is defining the offside line ([18–20]). In our view, these findings are compatible with the heuristic we are suggesting, although they do not reveal the critical stimulus that triggers offside decisions.

We addressed these issues in four studies on publicly available data from matches from the German Bundesliga. In Study 1, we directly compared the frequency of offside decisions in the so-called "Revierderby", that is, in matches between Borussia Dortmund und F.C. Schalke 04 (between 2013 and 2021). In Study 2, we then analyzed the frequency of offside decisions in matches between Borussia Dortmund and all other teams (except Schalke 04), and in matches between Schalke 04 and all other teams (except Borussia Dortmund) in season 2018/2019 of the German Bundesliga. In Studies 3 and 4, we attempted to replicate the pattern of results of Study 2 for a different season (Study 3) and different teams (Study 4).

All studies that are reported in this paper have a correlational, and not an experimental design. Hence, our studies can demonstrate a correlation between dress color (yellow vs. blue) and the number of offside decisions (low vs. high), but our studies cannot provide evidence for a causal link between dress color and (the number of) offside decisions. Hence, the primary purpose of our study is to provide empirical evidence for the existence of a correlation between dress color and (the number of) offside decisions in archival data, and to show that this correlation generalizes across time (Studies 2 and 3) and teams (Studies 2 and 4). The secondary purpose of our study is to propose a plausible (causal) explanation for the observed correlation, in terms of differences of figure-background contrast for yellow and blue shirts in a typical football stadium, and to provide indirect evidence for our account. Yet, we cannot provide direct experimental evidence for this account because we have no means of measuring figure-background contrasts in the available data.

Finally, the Video-Assistant Referee (VAR) was introduced into the German Bundesliga in season 2017/18, and one might think that the VAR would eliminate all biases and errors in offside decisions. In fact, the VAR is supposed to check a non-offside decision when a goal was scored from a potential offside situation, and overrule erroneous judgments of the AR. Studies 2 and 3 involve only games during which the VAR could have corrected possible errors of the AR, and therefore these studies provide a test of whether the VAR prevents color-related biases in offside judgments.

## Study 1

In our first study, we directly compared the frequency of offside decisions in the so-called "Revierderby", that is, in matches between Borussia Dortmund and FC Schalke 04. If the findings from our previous laboratory study, which suggested that the outfit of Schalke 04 (i.e.,

blue shirts and white shorts) was more salient than the outfit of Borussia Dortmund (i.e., yellow shirts and black shorts), generalized to real matches, we should observe more offside decisions against Schalke 04 than against Borussia Dortmund. A possible reason for that pattern would be that ARs more often miss actual offside positions from players of Borussia Dortmund than from players of Schalke 04.

## Methods

**Design and data analysis.** Our study rests on a simple one-factorial (within-subjects) design with attacking team (i.e., Borussia Dortmund or Schalke 04) as the independent variable. The design is equivalent to a within-subjects design because each match gives a pair of measures in both conditions. As the dependent variable, we analyzed and reported both the absolute number of offside decisions against each team, and an offside score, which was obtained by dividing the number of offside decisions against team x (in a particular match) by the number of shots by team x on the opponents' goal. Comparing the sum of offside decisions against each team provides an unbiased measure only if both teams are playing similarly offensive, that is, if both teams are creating a similar number of opportunities for scoring goals. If, however, team A is more dominant or playing riskier than team B, then team A will more often attack the goal of team B, and consequently the number of potential offside situations will be higher for team A than for team B. Hence, to obtain an estimate for the number of offside situations that is less affected by team dominance or playing style (offensive, defensive), we divided the number of offside decisions for team x by the number of shots (on the opponents' goal) by team x. Number of shots is a good estimate for team dominance and/or offensive playing style because previous research has shown that the number of total shots and the number of shots on target are among the best predictors of success in soccer (e.g., [21–23]). We used two-tailed $t$ tests for testing our hypotheses on the data, and we provide Cohen's $d_z$ as an effect-size measure. We performed our statistical analyses with the software „Jamovi", version 1.6.23.0 (www.jamovi.org). A formula for the effect size measure $d_z$ can be found in [24].

**Data source and materials.** We searched freely accessible internet databases for the required data (i.e., offside decisions against each team, number of shots on opponents' goal) for the matches between Borussia Dortmund and Schalke 04 in the German Bundesliga. We found the required data in the internet database of the "kicker" magazine (www.kicker.de), a German soccer magazine. Unfortunately, the kicker database provides the number of offside decisions against each team only from season 2013/14 onward. Hence, from the kicker database (and other freely accessible databases as well), we could only get information about the 16 most recent matches between Dortmund and Schalke. In all matches in our sample, the players of Borussia Dortmund wore yellow shirts and black shorts, whereas the players of Schalke 04 wore blue shirts and white shorts. We secured this fact by checking photographs from each match that were freely available on the internet.

In order to substantiate our hypothesis that differences in figure-background contrast are related to differences in offside decisions for differently colored players, it would have been desirable to determine the luminance contrasts for the teams and matches included in our study. This, however, was impossible because the background in modern soccer arenas is neither homogeneous nor static. When an AR is watching players, the background is a complex mixture of the playing field, spectators, flags, static advertisement boards, and electronic screens for presenting dynamic advertisements (e.g., [25]). Hence, the background of players in potential offside positions in our studies varied as a function of different variables such as (a) the perspective of the AR, (b) the position of the player on the field (with regard to AR and background), (c) the colors on static advertisement boards, and (d) the permanently changing

colors on the advertisement screens. Pictures with representative samples of these conditions were not available to us.

The video-assistant referee (VAR) was introduced in the German Bundesliga in season 2017/2018, and therefore might have affected offside judgments in some of the matches included in Study 1. We do not know how often this occurred because this information is not provided in the kicker database. It must be stressed, however, that the VAR would work against the expected effects of outfit color (or figure-ground contrast) on offside decisions.

## Results and discussion

There were, on average, more offside decisions against Schalke 04 ($M = 2.750$, $SD = 1.770$) than against Borussia Dortmund ($M = 1.313$, $SD = 1.078$), $t(15) = 3.216$, $p = .006$, $d_z = 0.804$. Numerically, Borussia Dortmund made more shots on the opponents' goal ($M = 13.000$, $SD = 6.000$) than Schalke 04 ($M = 9.250$, $SD = 3.235$), but this difference was not significant, $t(15) = 1.806$, $p = .091$, $d_z = 0.451$. Finally, the offside score (offside decisions against team/shots on opponents' goal) was higher for Schalke 04 ($M = 0.340$, $SD = 0.261$) than for Borussia Dortmund ($M = 0.113$, $SD = 0.098$), $t(15) = 3.661$, $p = .002$, $dz = 0.916$. The results of this analysis, therefore, clearly revealed more offside decisions against Schalke 04 than against Borussia Dortmund in matches between these two clubs in the German Bundesliga between 2013 and 2021.

## Study 2

Results of Study 1 revealed more offside decisions against Schalke 04 than against Borussia Dortmund when these two teams played against each other. We now wanted to know whether this finding was specific for this match, called the "Revierderby" in Germany, or whether the effect would also occur in matches against other teams. Therefore, in Study 2, we compared offside decisions in all matches of Borussia Dortmund and Schalke 04 against the 16 other teams in season 2018/19 of the German Bundesliga. We chose this particular season because it was the most recent season still played under „normal" conditions, that is, before the Corona pandemic. If the finding from Study 1 would generalize to other matches as well, we should observe more offside decisions against Schalke 04 than against their opponents (excluding Borussia Dortmund) and/or less offside decisions against Borussia Dortmund than against their opponents (excluding Schalke 04). Moreover, it seems reasonable to assume that the outfits of all other teams have, on average, intermediate figure-background contrast. Hence, Study 2 can also investigate whether the outfit of Borussia Dortmund has a salience (or figure-background contrast) that is below average, and whether the outfit of Schalke 04 has a salience (or figure-background contrast) that is above average. Finally, the VAR could have checked the offside judgments of the AR, and overrule erroneous judgments in all matches involved in Study 2. Hence, Study 2 provides a test of whether the VAR removes color-related biases in offside judgments.

### Methods

**Design and data analysis.** Study 2 rested on a three-factorial design with the factors Match Type (level 1: Borussia Dortmund vs. team x; level 2: Schalke 04 vs. team x), Place of Match (level 1: Borussia Dortmund or Schalke 04 played at home; level 2: Borussia Dortmund or Schalke 04 played away), and Team in Focus (level 1: Borussia Dortmund or Schalke 04; level 2: Other team from German Bundesliga). Note that the critical finding involves a two-way interaction of Match Type × Team in Focus: Fewer offside judgments for Borussia Dortmund than for their opponent in matches of Borussia Dortmund, and/or more offside judgments for Schalke 04 than for their opponent in matches of Schalke 04. If this two-way

interaction was significant, we planned to conduct two pair-wise comparisons for finding the source of the interaction. The pair-wise comparisons were performed in a one-tailed manner because we had specific hypotheses for each comparison. The primary dependent variable was the "offside score" (i.e., the number of offside decisions against team x / number of shots by team x on the opponents' goal), but we also report the results for absolute number of offside judgments as dependent variable.

**Data source and materials.** We garnered our data from the kicker database, which provided all required data about the matches from season 2018/19 in the German Bundesliga. It should be noted that both Borussia Dortmund and Schalke 04 had different outfits for home and away games in that season. The home outfit of Borussia Dortmund consisted of yellow shirts and black shorts. They wore this outfit in all home matches and in 15 away games (32 games in total). The away outfit of Borussia Dortmund consisted of black shirts and yellow shorts; they wore this outfit in 2 matches only. The home outfit of Schalke 04 consisted of blue shirts and white shorts. They wore this outfit in all home matches and in five away games (22 matches in total). The away outfit of Schalke 04 consisted of silver shirts and blue or silver shorts (5 matches), or of green shirts and blue or green shorts (7 matches).

## Results and discussion

The three-way ANOVA (Match Type × Place of Match × Team in Focus), with the number of offside decisions as the dependent variable, revealed a significant two-way interaction of Match Type and Team in Focus, $F(1, 60) = 7.038$, $MSE = 2.349$, $p = 0.010$, $partial\ \eta^2 = .105$. This interaction resulted from numerically less offside decisions against Borussia Dortmund ($M = 2.000$, $SD = 1.270$) than against their opponent ($M = 2.406$, $SD = 1.757$), $t(31) = -1.098$, $p = .140$, $d_z = -0.194$, and more offside decisions against Schalke 04 ($M = 2.375$, $SD = 1.737$) than against their opponents ($M = 1.344$, $SD = 1.260$), $W = 271.000$, $p = .014$, $r_b = .544$. If only those matches were considered, in which Borussia Dortmund wore yellow shirts, there were still numerically less offside decisions against Borussia Dortmund than against their opponent, $t(29) = -1.191$, $p = .122$, $d_z = -0.218$. Finally, if only those matches were considered, in which Schalke 04 wore blue shirts, there were still significantly more offside decisions against Schalke than against their opponent, $W = 135.000$, $p = .005$, $r_b = .765$.

The three-way ANOVA, with offside scores as dependent variable, also revealed a significant two-way interaction of Match Type and Team in Focus, $F(1, 60) = 13.141$, $MSE = 0.036$, $p < .001$, $partial\ \eta^2 = .180$. This interaction resulted from the fact that Borussia Dortmund had numerically lower offside scores ($M = 0.171$, $SD = 0.133$) than their opponents ($M = 0.288$, $SD = 0.236$), $W = 181.000$, $p = .062$, $r_b = -.314$, whereas Schalke 04 had higher offside scores than their opponents, $W = 390.500$, $p < .001$, $r_b = .680$. The marginal means are depicted in **Fig 1**. If only those matches were considered, in which Borussia Dortmund wore yellow shirts, their offside scores were significantly lower than those of their opponents, $W = 143.000$, $p = .033$, $r_b = -.385$. Finally, if only those matches were included, in which Schalke 04 wore blue shirts, their offside scores were still significantly higher than those of their opponents, $W = 163.000$, $p = .003$, $r_b = .716$.

The results of Study 2 showed that the findings from Experiment 1, more offside decisions against Schalke 04 than against Borussia Dortmund, generalized to matches against other teams. In season 2018/19 of the German Bundesliga, there were less offside decisions against Borussia Dortmund than against their opponents, and more offside decisions against Schalke 04 than against their opponents. These findings suggest that the salience of Borussia Dortmund's outfit is below average, whereas the salience of the outfit of Schalke 04 is above average.

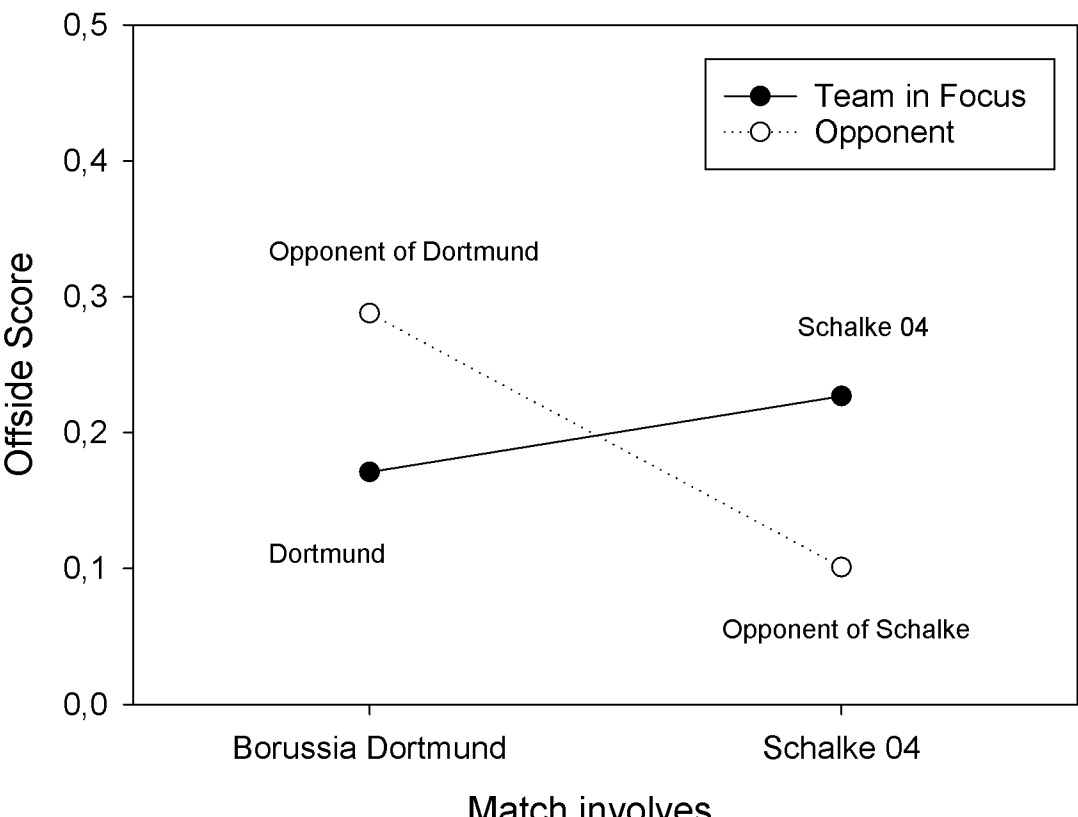

**Fig 1. Offside scores observed in Study 2 for Borussia Dortmund and their opponents, and Schalke 04 and their opponents, in matches of the German Bundesliga in season 2018/19.**

## Study 3

It is important to show that the pattern of results observed in Study 2 is not confined to a single season. Therefore, the main purpose of Study 3 was to replicate the pattern of results of Study 2 for the same teams (i.e., Borussia Dortmund and Schalke 04), but for a different season (i.e., 2017/2018). Again, the VAR could have checked the offside judgments of the AR, and overrule erroneous judgments in all matches involved in Study 3. Yet, the results of Study 2 suggest that the VAR cannot remove color-related biases in offside judgments, and we expect that Study 3 will replicate the pattern.

### Methods

**Design and data analysis.** Study 3 rested on the same design as Study 2. Again, the critical finding involves a two-way interaction of Match Type × Team in Focus: Fewer offside judgments for Borussia Dortmund than for their opponent in matches of Borussia Dortmund, and/or more offside judgments for Schalke 04 than for their opponent in matches of Schalke 04. If this two-way interaction was significant, we planned to conduct pair-wise comparisons for finding the source of the interaction. The primary dependent variable was the "offside score", but we also report the results for absolute number of offside judgments as dependent variable.

**Data source and materials.** We garnered our data from the kicker database. It should be noted that Borussia Dortmund and Schalke 04 had different outfits for home and away games

in season 2017/2018. The home outfit of Borussia Dortmund consisted of yellow shirts and black shorts. They wore this outfit in all home matches and in 5 away games (22 games in total). The away outfit of Borussia Dortmund consisted of yellow shirts and yellow shorts (6 matches) or black shirts and yellow shorts (6 matches). Hence, Borussia Dortmund wore yellow shirts in most matches (i.e., 28). The home outfit of Schalke 04 consisted of blue shirts and white shorts. They wore this outfit in all home matches and in three away games (20 matches). The away outfit of Schalke 04 consisted of blue shirts and blue shorts (8 matches), white shirts and blue shorts (3 matches), or white shirts and white shorts (3 matches). Hence, Schalke 04 wore blue shirts in most matches (i.e., 28).

## Results and discussion

The three-way ANOVA (Match Type × Place of Match × Team in Focus), with the number of offside decisions as the dependent variable, revealed a significant two-way interaction of Match Type and Team in Focus, $F(1, 60) = 7.038$, $MSE = 2.349$, $p = 0.010$, $partial \; \eta^2 = .105$. This interaction resulted from less offside decisions against Borussia Dortmund ($M = 2.531$, $SD = 1.951$) than against their opponents ($M = 3.500$, $SD = 2.229$), $t(31) = -1.747$, $p = .045$, $d_z = -0.309$, and more offside decisions against Schalke 04 ($M = 2.375$, $SD = 1.809$) than against their opponents ($M = 1.406$, $SD = 1.160$), $t(31) = 2.871$, $p = .004$, $d_z = 0.507$. If only those matches were considered, in which Borussia Dortmund wore yellow shirts, we observed numerically less offside decisions against Borussia Dortmund than against their opponents, $t(25) = -1.400$, $p = .087$, $d_z = -0.275$. Finally, if only those matches were considered, in which Schalke 04 wore blue shirts, we observed significantly more offside decisions against Schalke than against their opponent, $t(25) = 2.481$, $p = .010$, $d_z = 0.487$.

The three-way ANOVA with offside scores as dependent variable revealed a significant main effect of Match Type, $F(1, 60) = 7.141$, $MSE = 0.060$, $p = .010$, $partial \; \eta^2 = .106$, and a significant two-way interaction of Match Type and Team in Focus, $F(1, 60) = 4.717$, $MSE = 0.069$, $p = .034$, $partial \; \eta^2 = .073$. The main effect reflected a larger offside score in matches of Borussia Dortmund ($M = 0.300$) than in matches of Schalke 04 ($M = 0.184$). The interaction resulted from the fact that Borussia Dortmund ($M = 0.221$, $SD = 0.342$) had significantly smaller offside scores than their opponents ($M = 0.379$, $SD = 0.288$), $W = 108.000$, $p = .003$, $r_b = -.565$, whereas Schalke 04 ($M = 0.206$, $SD = 0.164$) had numerically larger offside scores than their opponents ($M = 0.162$, $SD = 0.160$), $W = 300.000$, $p = .084$, $r_b = .290$. The marginal means are depicted in **Fig 2**. If only those matches were considered, in which Borussia Dortmund wore yellow shirts, their offside scores were still significantly smaller than those of their opponents, $t(25) = -3.122$, $p = .002$, $d_z = -0.612$. Finally, if only those matches were included, in which Schalke 04 wore blue shirts, their offside scores were numerically larger than those of their opponents, $W = 196.000$, $p = .187$, $r_b = .206$. In summary, the results for season 2017/2018 (Study 3) replicate the results from season 2018/2019 (Study 2).

## Study 4

Studies 2 and 3 consistently showed less offside decisions against Borussia Dortmund and more offside decisions against Schalke 04 during two consecutive seasons. At first sight, these findings may support our hypothesis that yellow shirts are related to lower numbers of offside decisions, whereas blue shirts are related to higher numbers of offside decisions. Yet, until now, we have found this pattern for only two teams, and therefore its possible that the different numbers of offside decisions are not related to team colors, but to other team characteristics, such as ability or strategy. For example, it is possible that Schalke 04 more often played long passes to their forwards than Borussia Dortmund, which may have provoked more offside

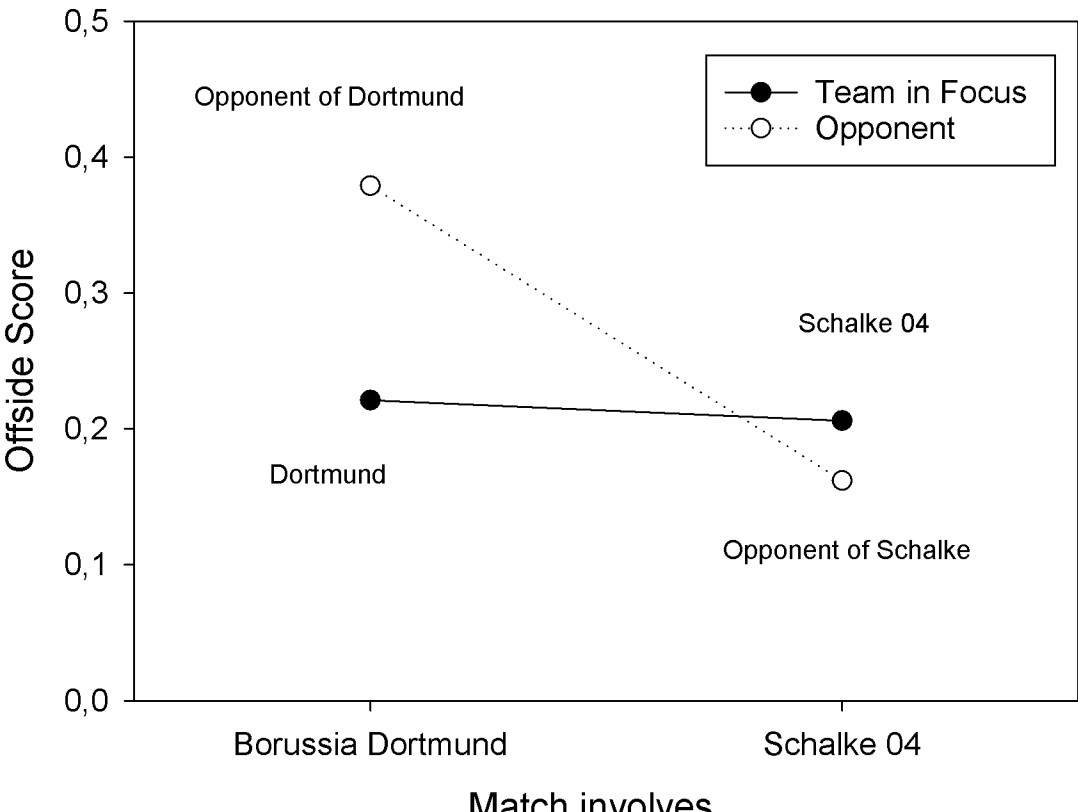

**Fig 2. Offside scores observed in Study 3 for Borussia Dortmund and their opponents, and Schalke 04 and their opponents, in matches of the German Bundesliga in season 2017/18.**

situations against the former team. One way of testing whether the relationship between dress color and offside decisions is specific for particular teams is to test whether a similar relationship occurs for other teams that are wearing the same colors, but differ in players, ability and playing style. Hence, in Study 4, we tested whether the pattern, observed for Borussia Dortmund and Schalke 04 in our previous studies, would also occur for Dynamo Dresden and VfL Bochum. These are two German soccer clubs that played in the second German Bundesliga during season 2018/2019. Importantly, players of Dynamo Dresden wear yellow shirts and black shorts, like Borussia Dortmund, whereas players of VfL Bochum wear blue shirts and white shorts, similar to Schalke 04. Observing less offside decisions against Dynamo Dresden and more offside decisions against VfL Bochum, when compared to all other teams, would replicate the findings from Study 2 and demonstrate that the pattern does not depend on a particular team with its particular players and playing habits.

## Methods

**Design and data analysis.**    The design of Study 4 was the same as that of studies 2 and 3.

**Data source and materials.**    As for our previous studies, we garnered our data from the kicker database. It should be noted that both Dynamo Dresden and VfL Bochum had different outfits for home and away games in season 2018/19. The home outfit of Dynamo Dresden consisted of yellow shirts and black shorts. They wore this outfit in all home matches and in 13 away games (30 games in total). The away outfit of Dynamo Dresden consisted of white shirts

with red stripes, and red shorts (4 matches). Hence, Dynamo Dresden wore yellow shirts in most matches (i.e., 30). The home outfit of VfL Bochum consisted of blue shirts and white shorts. They wore this outfit in all home matches and in in nine away games (26 matches). The away outfit of VfL Bochum consisted of green shirts and green shorts (5 matches), or gray shirts and black shorts (3 matches). Hence, VfL Bochum wore blue shirts in most matches (i.e., 26). The VAR could not have affected the results of Study 4 because he was introduced to the second Bundesliga in a later season (i.e., 2019/2020).

## Results and discussion

The three-way ANOVA (Match Type × Place of Match × Team in Focus), with the number of offside decisions as the dependent variable, revealed (only) a significant two-way interaction of Match Type and Team in Focus, $F(1, 64) = 5.755$, $MSE = 3.194$, $p = 0.019$, partial $\eta^2 = .083$. This interaction resulted from less offside decisions against Dynamo Dresden ($M = 2.118$, $SD = 1.472$) than against their opponents ($M = 3.000$, $SD = 2.103$), $t(33) = -1.886$, $p = .034$, $d_z = -0.323$, and numerically more offside decisions against VfL Bochum ($M = 2.529$, $SD = 1.656$) than against their opponents ($M = 1.941$, $SD = 1.556$), $t(33) = 1.452$, $p = .078$, $d_z = 0.249$. If only those matches were considered, in which Dynamo Dresden wore yellow shirts, we observed numerically less offside decisions against Dresden than against their opponents, $t(29) = -1.638$, $p = .056$, $d_z = -0.299$. Finally, if only those matches were considered, in which VfL Bochum wore blue shirts, we observed numerically more offside decisions against Bochum than against their opponent, $t(25) = 1.681$, $p = .053$, $d_z = 0.330$.

The three-way ANOVA with offside scores as dependent variable revealed (only) a significant two-way interaction of Match Type and Team in Focus, $F(1, 64) = 8.278$, $MSE = 0.032$, $p = 0.005$, partial $\eta^2 = .115$. The interaction resulted from the fact that Dynamo Dresden ($M = 0.187$, $SD = 0.151$) had numerically smaller offside scores than their opponents ($M = 0.259$, $SD = 0.231$), $t(33) = -1.505$, $p = .071$, $d_z = -.258$, whereas VfL Bochum ($M = 0.233$, $SD = 0.199$) had significantly larger offside scores than their opponents ($M = 0.127$, $SD = 0.101$), $t(33) = 2.804$, $p = .004$, $r_b = .481$. The marginal means are depicted in **Fig 3**. If only those matches were considered, in which Dynamo Dresden wore yellow shirts, their offside scores were significantly smaller than those of their opponents, $t(29) = -1.792$, $p = .042$, $d_z = -0.327$. Finally, if only those matches were included, in which VfL Bochum wore blue shirts, their offside scores were significantly larger than those of their opponents, $t(25) = 2.501$, $p = .010$, $d_z = .490$.

The results of Study 4 showed that the finding of less offside decisions against a team wearing a yellow/black outfit, and more offside decisions against a team wearing a blue/white outfit, is not confined to Dortmund and Schalke, but can also be observed for different teams (i.e., Dynamo Dresden and VfL Bochum). This finding is important because it constrains the number of variables that may account for the observed difference.

## General discussion

The present study tested the hypothesis that different shirt colors, which are presumably associated with different figure-background contrast, affect the number of offside judgments against a particular team. In a previous laboratory study, Wühr et al. [10] observed more offside judgments for forwards wearing the outfit of Schalke 04 (blue shirts, white shorts) than for forwards wearing the outfit of Borussia Dortmund (yellow shirts, black shorts), when the figure-background luminance contrast was higher for Schalke 04 than for Borussia Dortmund. Here, we asked whether a similar pattern would occur in real matches of the German Bundesliga. In Study 1, we directly compared offside scores (i.e., number of offside judgments against

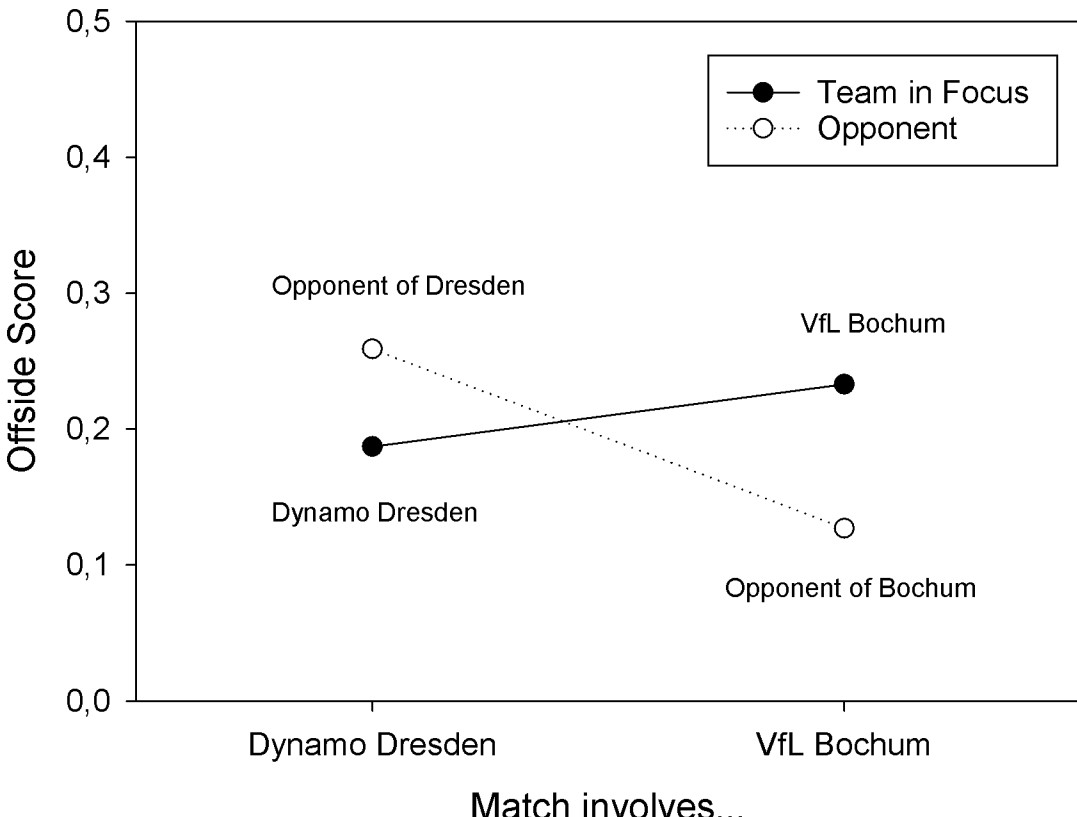

**Fig 3. Offside scores observed in Study 4 for Dynamo Dresden and their opponents, and VfL Bochum and their opponents, in matches of the second German Bundesliga in season 2018/19.**

a team divided by the number of shots by this team on the opponents' goal) in the 16 most recent matches between Schalke 04 and Borussia Dortmund. Results showed a higher offside score for Schalke 04 than for Borussia Dortmund. In Study 2, we separately compared offside scores of Schalke 04 and Borussia Dortmund in their games against all other teams of the German Bundesliga during season 2018/19. Results showed a lower offside score for Borussia Dortmund against all other teams, and a higher offside score for Schalke 04 against all other teams. In Studies 3 and 4, we showed that the pattern observed in Study 2 generalizes across seasons and teams. In Study 3, we replicated the pattern for Dortmund and Schalke in season 2017/2018. Finally, in Study 4, we replicated the pattern for another team wearing a yellow-black outfit (i.e., Dynamo Dresden), and another team wearing a blue-white outfit (i.e., VfL Bochum). In summary, the results of our studies provide strong evidence for a correlation between shirt color of a team, and the number of offside decisions against this team: yellow shirts are associated with lower numbers of offside decisions, whereas blue shirts are associated with larger numbers of offside decisions.

## Shirt color and figure-background contrast

Our studies provide strong evidence for a correlation between shirt color and the number of offside decisions. A plausible account for this finding relates the differences between shirt colors to differences in their figure-background luminance contrast. Hence, to account for the results of Study 1, we would have to assume that figure-background luminance contrast would

be higher for Schalke 04 than for Borussia Dortmund, although Borussia Dortmund is wearing the brighter shirt. Yet, if background luminance is closer to the luminance of the Borussia Dortmund dress than to the luminance of the Schalke 04 dress, contrast would be higher for Schalke 04 than for Borussia Dortmund. Similarly, to account for the results of Studies 2–4, we would have to assume that figure-background contrast for yellow shirts is lower than the average contrast of the outfits of all other teams, whereas the figure-background contrast for blue shirts is higher than the average contrast of the outfits of all other teams.

As an interesting theoretical implication, the observed correlation between shirt colors and offside decisions may inform us about the cognitive heuristics that ARs are using for spotting offside positions. It seems plausible that ARs, in order to detect players in offside positions, search among the players that are near the offside line for the player that is closest to the goal under attack. If this player is a forward, the response will be "offside;" if this player is a defender, the response will be "no offside". Because attention is biased towards higher-contrast stimuli (e.g., [16,17]), ARs will more often "find" the higher-contrast player (in a blue shirt) than the lower-contrast player (in a yellow shirt), and therefore the bias would increase offside judgments against Schalke 04, and decrease offside judgments against Borussia Dortmund. Of course, this account must be taken with caution because we have not measured figure-background contrast in the matches included in our studies, but we know that higher figure-background contrast can increase the number of offside decisions against the more salient team from previous laboratory studies ([10]). Hence, we can provide indirect empirical evidence for our account.

Having shown that blue shirts and white shorts co-occur with an above-average number of offside decisions, whereas yellow shirts and black shorts co-occur with a below-average number of offside decisions, does not lead us to claim that these findings will necessarily generalize to every other team wearing similar colors. We believe that figure-background contrast plays an important role for these effects, and this contrast does not only depend upon the figures (i.e., players) but also on other variables including the background, illumination conditions, the viewing angle of the observer (referee), and probably more. Only if most of these variables take sufficiently similar values, we would expect similar effects of dress color on offside decisions. In other words, it is not our intention to show that yellow shirts will always decrease offside decisions, or that blue shirts will always increase offside decisions, but to show that shirt colors can affect figure-background contrast, which in turn can affect offside decisions of (assistant) referees.

## Alternative accounts?

Several characteristics of a team, besides outfit color, and several characteristics of a match situation could (also) affect the number of offside decisions against that team. Among the team characteristics are the individual abilities of the players, and team strategy. Concerning player ability, it seems likely that less able players more often run into offside positions than better players. This would imply that the blue-white teams in our studies consisted of players that were less able than average, whereas the yellow-black teams in our studies consisted of players that were better than average. This seems implausible, given the fact that Schalke finished better than Borussia Dortmund in season 2018/2019, and VfL Bochum finished better than Dynamo Dresden in the same year. Concerning team strategy, it is possible that teams have a strategy that provokes more offside situations, such as playing long passes from behind towards their forwards. This is also implausible because, in order to explain the observed pattern of results, both yellow teams (Dortmund and Dresden) should have used strategies that lead to a less-than-average number of offside situations, whereas both blue teams (Schalke and

Bochum) should have used strategies that lead to a more -than-average number of offside situations. Moreover, the fact that we did not observe significant differences in the number of offside situations in home versus away matches can also be viewed as evidence against a role of team strategy, because many teams play differently when at home or when away (e.g., [26,27]).

Among the features of a match situation that could also affect the number of offside situations are the opposing team and the referees (i.e., AR's). In order to account for the observed pattern, the opposing teams must have played differently against yellow or blue teams. In particular, the opponents must have played against the yellow teams in a way that creates a less-than-average number of offside situations, whereas the same teams must have played against the blue teams in a way that produces a more-than-average number of offside situations. Again, this hypothesis is implausible. Finally, to account for the observed pattern, the referees should have assigned a less-than-average number of offside decisions against the yellow teams in our study, whereas the referees should have assigned a more-than-average number of offside decisions against the blue teams in our study for reasons that are unrelated to color or player visibility. We cannot imagine a non-perceptual variable that would lead many different referees to treat yellow and blue teams differently in offside situations. In summary, it seems hard to imagine a variable, which is not related to outfit color and player visibility, but could equally well explain the observed difference in offside decisions between yellow and blue teams.

## Further implications

We would like to stress two implications of our findings. First, an impact of player salience, due to differences of figure-background luminance contrast, would constitute a team-specific and durable source of errors in offside judgments, and may probably also affect other referee decisions. This effect might produce an (inadvertent) advantage for teams wearing dresses of relatively low salience, and an (inadvertent) disadvantage for teams wearing dresses of relatively high salience. The good news is that these effects are avoidable by changing the luminance, and hence the relative figure-background contrast, of team outfits. In fact, using dresses that differ only in color, but not in luminance, would eliminate any effects of luminance-based contrast effects on offside judgments.

Second, the fact that an effect, which was originally observed in a simple version of the offside-judgments task with laymen participants [10], could also be found in data from real soccer games at the highest professional level in Germany, suggests that our laboratory task captures some important features of offside judgments in real games.

Third, we observed a color-related bias in offside judgments of professional referees in two seasons of the German Bundesliga (i.e., in Studies 2 and 3), when the VAR was already present. Hence, the results of our studies demonstrate that the VAR cannot remove color-related biases in offside judgments, which may seem surprising to some readers. There are two possible ways of how the VAR might fail to remove the bias. First, it is possible that the color-related bias affects offside judgments of the AR only in perceptually difficult situations (i.e., when the forward and the defender are almost level), where the AR does not intervene. Second, it is also possible that the color-related bias can also affect offside judgments of the VAR, which may also occur only under difficult viewing conditions.

## Strength and limitations

Because we have used only publicly accessible data for the present studies, everyone can check and verify our results. Moreover, the observed relationships between outfit colors and number of offside decisions are robust and generalizable across seasons and teams (clubs). Finally, despite the restricted number of observations per study (i.e., a maximum of 34 matches per

team), the observed effects were relatively strong. In fact, the effect size of most pair-wise comparisons was larger than $d > .40$, which is considered a lower limit for the practical significance of empirical findings (e.g., [28]).

On the downside, the present data "only" demonstrate a correlational relationship between shirt colors and the number of offside decisions, but they cannot demonstrate a causal effect of shirt color (or figure-background contrast) on offside decisions. In other words, we cannot exclude a possible confounding of shirt color with other variables that might also affect the number of offside decisions. Obtaining realistic measures of background contrast during real matches in football arenas would be a useful topic for future research. Alternatively, future research could also address the impact of figure-background contrast on offside judgments in well-controlled laboratory studies, and we have already begun to tackle this issue.

## Author Contributions

**Conceptualization:** Peter Wühr, Daniel Memmert.

**Data curation:** Peter Wühr.

**Formal analysis:** Peter Wühr.

**Investigation:** Peter Wühr.

**Writing – original draft:** Peter Wühr, Daniel Memmert.

**Writing – review & editing:** Peter Wühr, Daniel Memmert.

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
