## [Decision Letter · Decision Letter 0]

26 Dec 2022

PONE-D-22-32096A color-related bias in offside judgments in professional soccer: A matter of shirt-background contrast?PLOS ONE

Dear Dr. Wuehr,

Thank you for submitting your manuscript to PLOS ONE. After careful consideration, we feel that it has merit but does not fully meet PLOS ONE’s publication criteria as it currently stands. Therefore, we invite you to submit a revised version of the manuscript that addresses the points raised during the review process.

We look forward to receiving your revised manuscript.

Kind regards,

Yair Galily

Academic Editor

PLOS ONE

Journal Requirements:

Reviewers' comments:

Reviewer's Responses to Questions

**Comments to the Author**

1. Is the manuscript technically sound, and do the data support the conclusions?

Reviewer #1: No

Reviewer #2: Yes

Reviewer #3: Yes

2. Has the statistical analysis been performed appropriately and rigorously? 

Reviewer #1: No

Reviewer #2: Yes

Reviewer #3: Yes

3. Have the authors made all data underlying the findings in their manuscript fully available?

Reviewer #1: No

Reviewer #2: Yes

Reviewer #3: Yes

4. Is the manuscript presented in an intelligible fashion and written in standard English?

Reviewer #1: Yes

Reviewer #2: Yes

Reviewer #3: Yes

5. Review Comments to the Author

Reviewer #1: Summary:

The paper investigates the impact of outfit colors on the frequency of offside judgments in

soccer. For that the authors investigated games between a team in yellow jersey and a team wearing blue jerseys. The authors claim that more offside judgments are made against teams of higher salience (blue), possibly induced by differences in figure-background contrast.

Evaluation

The main challenge of such a paper is to control for teams' abilities and styles of play. To do so, the authors use the number of offsides called against a team divided by the number of shots a team kicked towards the opposing team. This measure is based on a very strong assumption that having more offsides is a good measure of teams’ offensive abilities compared to the opponent. However, I don't think that this is enough (and correct). It is plausible to assume that lower ability teams would struggle against higher ability teams both defensively and offensively. For example, in a recent World Cup game between Brazil and South Korea that was very much one sided in favor of Brazil (4:1), Brazilians had more kicks (18:10), but not a single offside compared to five offsides of South Korea (see https://www.fifa.com/fifaplus/en/match-centre/match/17/255711/285073/400128133?country=NO&wtw-filter=ALL ). Why can't it be the same in your case where Borussia Dortmund turned out to be a much better team than their blue-colored opponents? Thus, in my opinion, your identification strategy to disentangle ability from bias is not valid.

The second problem relates to the selection of blue-colored teams. To the best of my knowledge, there are several teams that play with blue jerseys (Hoffenheim, Hertha Berlin, Paderborn, Bochum, etc.). Thus, it is not clear why did the authors chose the very specific teams. What I expected to see is the entire pool of games of blue-colored teams, rather than only a very selective sample.

Reviewer #2: The present paper deals with the association between jersey color and offside decisions in soccer. I found this to be a very interesting topic that may well warrant publication in PLoS ONE.

The paper is very well written and interesting to read. I appreciate that the authors carefully discuss the limitations of their correlational approach. I am also convinced that the authors’ approach (first, find an effect in the lab; second, show that it also exists in the real world) is important for sport-psychological research, as it allows combining the search for causal mechanisms with real-world applicability. That said, I see a couple of points that I would like to see addressed before I can potentially recommend the manuscript for publication.

1)

The authors’ reasoning rests heavily on their proposed search strategy for ARs (e.g., lines 139-142; lines 493-496). The authors seem to be aware of the relevance of this idea for their study. For example, they suggest that their findings “may inform us about the cognitive heuristics that ARs are using for spotting offside positions” (lines 492 / 493). However, I was wondering: Is there any evidence that this really is what ARs are doing? If so, presenting this evidence might considerably strengthen the present reasoning. If not, what are potential alternative strategies for ARs, and how do the present results fit these strategies, if at all?

2)

Can the authors please elaborate a bit on the proposed luminance contrast: What exactly constitutes this contrast according to their reasoning: Is it the forward’s jersey vs the other team’s jerseys; or the forward’s jersey vs the green of the pitch; or the forward’s jersey vs a mixture of other jerseys and the green of the pitch; or any other combination? I understand that the authors could not empirically determine the respective contrast (see their explanations in lines 216-227). However, what do they principally propose? What is the contrast supposed to consist of?

Furthermore, can the authors briefly explain somewhere what exactly a luminance contrast is and why it is different for the jerseys of Dortmund and Schalke?

3)

I appreciate the careful discussion of potential alternative explanations (starting with line 505). However, I believe the authors are a bit quick to dismiss the alternative explanation that the results might have something to do with the respective teams’ playing styles (lines 516-520). To me, it does not seem to be implausible that both Dortmund and Dresden might have used somewhat similar tactics, just like Schalke and Bochum might have used a similar one, but a different one from the tactic used by Dortmund and Dresden. Maybe the authors can elaborate a little on this question. I do not see this as being a major problem: As the authors note themselves, their data are correlational, so it is obvious that not all alternative explanations can be ruled out.

Furthermore, they might want to discuss that their results completely hinge on data from 4 teams. What does that mean for their generalizability?

Reviewer #3: Report:

A color-related bias in offside judgments in professional soccer: A matter of shirt background contrast?

Kudos to the authors of this “well baked” paper. The Introduction is characterized by a logical flow as it smoothly brings the reader to the problem at hand. The authors doing a good job elaborating the mechanisms that can mediate the correlation between color salience and offside decisions. The methodological part is thoroughly explained, the authors treat their results with caution and doing a long way to deal with alternative explanations. The Discussion is also well-developed and sufficient.

I have very a few minor comments:

1. line 252: space between 1 and would

2. line 281: It is better to continue with shirts and trousers instead of using shirts and shorts

3. line 417: The VAR is a system, so use it instead of he

4. line 458: Wühr et al. (2020) not 2021

5. The authors present the reader in the Introduction with the issue of correlation vs. causation, it seems appropriate to mention here the accepted term: confounding problem and/or endogeneity.

With appreciation for your meticulous work,

Reviewer

6. PLOS authors have the option to publish the peer review history of their article (what does this mean?). If published, this will include your full peer review and any attached files.

Reviewer #1: No

Reviewer #2: No

Reviewer #3: **Yes: **Elia Morgulev

---

## [Author Response · Author response to Decision Letter 0]

28 Feb 2023

Revision of ms PONE-D-22-32096 (A color-related bias in offside judgments in professional soccer: A matter of figure-background contrast?).

Responses to Reviewers' comments:

Reviewer #1:

Summary: The paper investigates the impact of outfit colors on the frequency of offside judgments in soccer. For that the authors investigated games between a team in yellow jersey and a team wearing blue jerseys. The authors claim that more offside judgments are made against teams of higher salience (blue), possibly induced by differences in figure-background contrast.

Evaluation: The main challenge of such a paper is to control for teams' abilities and styles of play. To do so, the authors use the number of offsides called against a team divided by the number of shots a team kicked towards the opposing team. This measure is based on a very strong assumption that having more offsides is a good measure of teams’ offensive abilities compared to the opponent. However, I don't think that this is enough (and correct). It is plausible to assume that lower ability teams would struggle against higher ability teams both defensively and offensively. For example, in a recent World Cup game between Brazil and South Korea that was very much one sided in favor of Brazil (4:1), Brazilians had more kicks (18:10), but not a single offside compared to five offsides of South Korea (see https://www.fifa.com/fifaplus/en/match-centre/match/17/255711/285073/400128133?country=NO&wtw-filter=ALL ). Why can't it be the same in your case where Borussia Dortmund turned out to be a much better team than their blue-colored opponents? Thus, in my opinion, your identification strategy to disentangle ability from bias is not valid.

Author response: First, Reviewer 1 is not convinced that our offside score does effectively reduce or remove the impact of playing style, or team ability, from the absolute number of offsides. We do not agree. First, we do not claim that having more offsides is a good measure of offensive abilities. What we are claiming is that shots on the opponents goal is a good measure of playing style (offensive vs. defensive), and this relationship is supported by empirical data. For example, studies of the home advantage have shown that home teams make more shots on the opponents’ goal than away teams (e.g., Armanatas & Pollard, 2014; Poulter, 2009), which is often attributed to the fact that home teams play more offensive than away teams. Moreover, there is also evidence for a positive correlation between number of shots on opponents’ goal and offsides, since both are positively correlated with success (i.e., points obtained in a season) in the Spanish La Liga (Souza et al., 2019). We believe these empirically established relationships justify our computation of the offside score as an attempt to reduce the impact of playing style on the number of offside decisions against a particular team. Of course, the correlations reflect probabilistic, and not deterministic, relationships between the variables. Therefore, there will be exceptions to the general rule, as the match between Brazil and South Korea mentioned by Reviewer 1. Second, Reviewer 1 suggests that the observed differences in offsides simply reflect the fact that Borussia Dortmund is a much better team than their blue-colored opponents. Again, we do not agree. Note that we do not only show that Borussia Dortmund receives less offside decisions than Schalke 04. We show that two different yellow teams (Dortmund, Dresden) have less offsides than all other teams in the competition, whereas two different blue teams (Schalke, Bochum) have more offsides than all other teams in the competition. Hence, to explain the full pattern, Reviewer 1 would have to assume that the yellow teams (Dortmund, Dresden) are much better than the rest, whereas Schalke and Bochum are much worse than the rest. Both assumptions are wrong. In season 2017/18 (study 3), Schalke 04 finished the Bundesliga on rank 2, whereas Dortmund finished on rank 4. Similarly, in season 2018/19 (study 4), Bochum finished the second Bundesliga on rank 11, whereas Dresden finished on rank 12. Hence, because the critical teams performed similarly in both studies, differences in ability cannot explain the differences in offsides.

The second problem relates to the selection of blue-colored teams. To the best of my knowledge, there are several teams that play with blue jerseys (Hoffenheim, Hertha Berlin, Paderborn, Bochum, etc.). Thus, it is not clear why did the authors chose the very specific teams. What I expected to see is the entire pool of games of blue-colored teams, rather than only a very selective sample.

Author response: We chose Schalke 04 for Studies 1-3 because we have investigated the match between Borussia Dortmund and Schalke 04 in our previous laboratory experiments (Wühr et al., 2020), which inspired the present research. For the present study 4, we needed teams with similar colors. We chose VfL Bochum as a team with blue-white colors because Bochum resides in the same area as Dortmund and Schalke, but this choice was admittedly arbitrary. Next, Reviewer 1 asks why we did not investigate the entire pool of games of blue-colored teams. The main reason is that we neither assume nor claim that blue-white garments will always co-occur with an above-average number of offside decisions, and we neither assume nor claim that yellow-black garments will always co-occur with a below-average number of offside decisions. The primary goal of our study, as described in the introduction, is to provide empirical evidence for the existence of a correlation between dress color and (the number of) offside decisions in archival data, and to show that this correlation generalizes across time (Studies 2 and 3) and teams (Studies 2 and 4). This goal was achieved with four teams. Yet, we do not claim these findings will necessarily generalize to every other team wearing similar colors. We believe that figure-background contrast plays an important role for these effects, and this contrast does not only depend upon the figures (i.e., players) but also on other variables including the background, illumination conditions, the viewing angle of the observer (referee), and probably more. Only if all these variables are sufficiently similar, we would expect similar effects of dress color on offside decisions. We have added a statement to the General Discussion (section “Shirt color and figure-background contrast”) in order to clarify the goals of our study.

Reviewer #2:

Summary: The present paper deals with the association between jersey color and offside decisions in soccer. I found this to be a very interesting topic that may well warrant publication in PLoS ONE. The paper is very well written and interesting to read. I appreciate that the authors carefully discuss the limitations of their correlational approach. I am also convinced that the authors’ approach (first, find an effect in the lab; second, show that it also exists in the real world) is important for sport-psychological research, as it allows combining the search for causal mechanisms with real-world applicability. That said, I see a couple of points that I would like to see addressed before I can potentially recommend the manuscript for publication.

(1) The authors’ reasoning rests heavily on their proposed search strategy for ARs (e.g., lines 139-142; lines 493-496). The authors seem to be aware of the relevance of this idea for their study. For example, they suggest that their findings “may inform us about the cognitive heuristics that ARs are using for spotting offside positions” (lines 492 / 493). However, I was wondering: Is there any evidence that this really is what ARs are doing? If so, presenting this evidence might considerably strengthen the present reasoning. If not, what are potential alternative strategies for ARs, and how do the present results fit these strategies, if at all?

Author response: This is an interesting question. In fact, as far as we know the literature, quite little is known about what ARs are really doing when judging offside. In other words, we do not yet know what stimulus or information is eventually triggering the offside decision in ARs. There are, however, a couple of studies that have investigated the visual (i.e., scanning and fixation) behavior of ARs, and the results of these studies might be helpful here. We have added a paragraph summarizing the results and implications of these studies to the introduction of the revised manuscript. In our view, the results of these studies support the plausibility of the heuristic we are suggesting in our paper. The heuristic would predict that ARs should mostly look at the offside line, and at players (forwards, defenders) that are located close to the offside line. These predictions fit the empirical results (e.g., Luis Del Campo et al., 2018; Luis Del Campo & Martin, 2020; Catteeuw et al., 2009). In summary, the studies revealed that ARs always never fixate the player that plays the ball to a forward, while most of the time the ARs fixate a point in the vicinity of the offside line, or the second-last defender that is defining the offside line. In our view, these findings are compatible with the heuristic we are suggesting, although they do not reveal the critical stimulus that triggers offside decisions.

(2) Can the authors please elaborate a bit on the proposed luminance contrast: What exactly constitutes this contrast according to their reasoning: Is it the forward’s jersey vs the other team’s jerseys; or the forward’s jersey vs the green of the pitch; or the forward’s jersey vs a mixture of other jerseys and the green of the pitch; or any other combination? I understand that the authors could not empirically determine the respective contrast (see their explanations in lines 216-227). However, what do they principally propose? What is the contrast supposed to consist of? Furthermore, can the authors briefly explain somewhere what exactly a luminance contrast is and why it is different for the jerseys of Dortmund and Schalke?

Author response: We admit that we did not make this issue clear enough in the previous version of our manuscript. Therefore, we added the following paragraph to the introduction: The term “contrast” refers to differences in luminance (or brightness) between two, typically adjacent, areas of the visual field (see, Alexander, Xie, & Derlacki, 1993, for measures of contrast). Luminance contrasts can occur between figure and ground, or between two figures, and therefore may affect both figure-ground segmentation, and visual object recognition (e.g., Legge, Rubin, & Luebker, 1987; Legge, 1993; Regan & Beverley, 1984). The larger the luminance difference (contrast) between figure and background, the easier the figure can be separated from the background and, thus, perceived. A luminance contrast can be positive, when the figure has higher luminance than the background, or negative, when the figure has lower luminance than the background. In the present context, we believe that the critical variable is the difference between the figure-background contrast of team 1 and the figure-background contrast of team 2. Other variables being equal, the figure with higher figure-ground contrast will be detected and processed more easily than the figure with lower figure-ground contrast. In our previous study, the blue shirts were easier perceivable than the yellow shirts because the darker blue shirts contrasted more strongly with the relatively bright background than the brighter yellow shirts. In other words: differences in figure-background contrast occur whenever (a) the garments of two teams differ in luminance and (b) the luminance of the background is not in between the luminance of the garments.

(3) I appreciate the careful discussion of potential alternative explanations (starting with line 505). However, I believe the authors are a bit quick to dismiss the alternative explanation that the results might have something to do with the respective teams’ playing styles (lines 516-520). To me, it does not seem to be implausible that both Dortmund and Dresden might have used somewhat similar tactics, just like Schalke and Bochum might have used a similar one, but a different one from the tactic used by Dortmund and Dresden. Maybe the authors can elaborate a little on this question. I do not see this as being a major problem: As the authors note themselves, their data are correlational, so it is obvious that not all alternative explanations can be ruled out.

Author response: Unfortunately, it is very difficult to compare playing styles or tactics between teams. As a rough approximation to the issue, we compared (a) shots of the four analyzed teams (i.e., Bochum, Schalke, Dortmund, Dresden) on the opponents’ goal, as a measure of offensive play, and we compared (b) shots of the opponents on the goal of the analyzed teams, as a measure of defensive play. If the same-colored teams (Dortmund/Dresden; Bochum/Schalke) behaved similarly, but differently from the other-colored teams, we should observe corresponding differences in the two dependent variables. In particular, the yellow teams should have made less shots on their opponents’ goals than the blue teams if the difference in offsides (lower offsides for yellow as compared to blue teams) reflected a more defensive playing style of yellow teams. This was not the case, F(3, 124) = 0.350, p = .789. In contrast, Dortmund had numerically more shots on the opponents goal than the other teams. Concerning defensive behaviors (or abilities), the yellow teams should have allowed their opponents more shots on their goal than blue teams if the difference in offsides (more offsides for opponents of yellow teams than for opponents of blue teams) reflected a more offensive playing style (or less defensive abilities) of yellow as compared to blue teams. This was not the case, too. There was a significant main effect of team, F(3, 124) = 7.098, p < .001, reflecting the fact that Dortmund conceded less shots on their own goal (i.e. higher defensive qualities) than the other teams. Hence, the results of these analyses do not support the hypothesis that the same-colored teams played similarly, but differently from the other-colored teams. We did not yet report the results of this analysis in the revised manuscript, but we might do that if the Reviewers or the Action Editor consider this as useful. 

(4) Furthermore, they might want to discuss that their results completely hinge on data from 4 teams. What does that mean for their generalizability? 

Author response: This is an important point (see also comment 2 by Reviewer 1). As we have already written in the introduction, the primary purpose of our study is to provide empirical evidence for the existence of a correlation between dress color and (the number of) offside decisions in archival data, and to show that this correlation generalizes across time (Studies 2 and 3) and teams (Studies 2 and 4). Having shown that blue shirts and white shorts co-occur with an above-average number of offside decisions, whereas yellow shirts and black shorts co-occur with a below-average number of offside decisions, does not lead us to claim these findings will necessarily generalize to every other team wearing similar colors. We believe that figure-background contrast plays an important role for these effects, and this contrast does not only depend upon the figures (i.e., players) but also on other variables including the background, illumination conditions, the viewing angle of the observer (referee), and probably more. Only if all these variables are sufficiently similar, we would expect similar effects of dress color on offside decisions. Or, in other words, it is not our intention to show that yellow shirts will always decrease offside decisions, or that blue shirts will always increase offside decisions, but to show that shirt colors can affect figure-background contrast, which in turn can affect offside decisions of (assistant) referees. We have also added a corresponding statement to the General Discussion (section “Shirt color and figure-background contrast”). 

Reviewer #3:

Report: A color-related bias in offside judgments in professional soccer: A matter of shirt background contrast?

Kudos to the authors of this “well baked” paper. The Introduction is characterized by a logical flow as it smoothly brings the reader to the problem at hand. The authors doing a good job elaborating the mechanisms that can mediate the correlation between color salience and offside decisions. The methodological part is thoroughly explained, the authors treat their results with caution and doing a long way to deal with alternative explanations. The Discussion is also well-developed and sufficient.

I have very a few minor comments:

1. line 252: space between 1 and would

Author response: done.

2. line 281: It is better to continue with shirts and trousers instead of using shirts and shorts

Author response: We decided to use the term “shorts” consistently throughout the manuscript, and to remove the term “trousers” instead. 

3. line 417: The VAR is a system, so use it instead of he

Author response: Doesn’t VAR mean “Video Assistant Referee” and, therefore, should be a person?

4. line 458: Wühr et al. (2020) not 2021

Author response: done

5. The authors present the reader in the Introduction with the issue of correlation vs. causation, it seems appropriate to mention here the accepted term: confounding problem and/or endogeneity.

Author response: We agree, and added a corresponding sentence to the final paragraph (on page 25). 

With appreciation for your meticulous work,

Reviewer 3

References

Alexander, K. R., Xie, W., & Derlacki, D. J. (1993). The effect of contrast polarity on letter identification. Vision Research, 33, 2491–2497. 

Armanatas, V., & Pollard, R. (2014). Home advantage in Greek football. European Journal of Sport Science, 14, 116-122. 

Catteeuw, P., Helsen, W., Gilis, B., Van Roie, E., & Wagemans, J. (2009). Visual scan patterns and decision-making skills of expert assistant referees in offside situations. Journal of Sport & Exercise Psychology, 31, 786–797. 

Legge, G. E., Rubin, G. S., & Luebker, A. (1987). Psychophysics of reading: V The role of contrast in normal vision. Vision Research, 27, 1165–1177. 

Legge, G. E. (1993). The role of contrast in reading: Normal and low vision. In R. M. Shapley & D. M.-K. Lam (Eds.), Contrast sensitivity. (pp. 269–287). The MIT Press. 

Luis Del Campo, V., & Morenas Martín, J. (2020). Influence of video speeds on visual behavior and decision-making of amateur assistant referees judging offside events. Frontiers in Psychology, 11. 

Luis Del Campo, V., Fariñas, A. C., Domínguez Márquez, F. J., & Martín, J. M. (2018). The influence of refereeing experiences judging offside actions in football. Psychology of Sport and Exercise, 37, 139–145. 

Poulter, D. R. (2009). Home advantage and player nationality in international club football. Journal of Sports Sciences, 27, 797-805. 

Regan, D., & Beverley, K. I. (1984). Figure–ground segregation by motion contrast and by luminance contrast. Journal of the Optical Society of America, A, Optics, Image & Science, 1, 433–442. 

Souza, D. B., López-Del Campo, R., Blanco-Pita, H., Resta, R., & Del Coso, J. (2019): A new paradigm to understand success in professional football: analysis of match statistics in La Liga for 8 complete seasons, International Journal of Performance Analysis in Sport, 19, 543-555.

---

## [Decision Letter · Decision Letter 1]

25 Apr 2023

A color-related bias in offside judgments in professional soccer: A matter of figure-background contrast?

PONE-D-22-32096R1

Dear Dr. Wuehr,

We’re pleased to inform you that your manuscript has been judged scientifically suitable for publication and will be formally accepted for publication once it meets all outstanding technical requirements.

Kind regards,

Goran Kuvačić, PhD

Academic Editor

PLOS ONE

Additional Editor Comments (optional):

Reviewers' comments:

Reviewer's Responses to Questions

**Comments to the Author**

1. If the authors have adequately addressed your comments raised in a previous round of review and you feel that this manuscript is now acceptable for publication, you may indicate that here to bypass the “Comments to the Author” section, enter your conflict of interest statement in the “Confidential to Editor” section, and submit your "Accept" recommendation.

Reviewer #2: All comments have been addressed

Reviewer #3: All comments have been addressed

2. Is the manuscript technically sound, and do the data support the conclusions?

Reviewer #2: Yes

Reviewer #3: Yes

3. Has the statistical analysis been performed appropriately and rigorously? 

Reviewer #2: Yes

Reviewer #3: Yes

4. Have the authors made all data underlying the findings in their manuscript fully available?

Reviewer #2: Yes

Reviewer #3: Yes

5. Is the manuscript presented in an intelligible fashion and written in standard English?

Reviewer #2: Yes

Reviewer #3: Yes

6. Review Comments to the Author

Reviewer #2: The authors of the present paper have extensively provided answers to my suggestions and questions regarding their first version of the paper and they have added substantial changes to their original manuscript. I congratulate the authors on their detailed answers and thorough revisions.

Reviewer #3: The authors adressed my comments and I am satisfied with the improvments. The manuscript is ready for poblication.

7. PLOS authors have the option to publish the peer review history of their article (what does this mean?). If published, this will include your full peer review and any attached files.

Reviewer #2: No

Reviewer #3: **Yes: **Elia Morgulev

<quillbot-extension-portal></quillbot-extension-portal>

---

## [Editor Report · Acceptance letter]

3 May 2023

PONE-D-22-32096R1 

A color-related bias in offside judgments in professional soccer: A matter of figure-background contrast? 

Dear Dr. Wühr:

I'm pleased to inform you that your manuscript has been deemed suitable for publication in PLOS ONE. Congratulations! Your manuscript is now with our production department. 

Kind regards, 

on behalf of

Dr. Goran Kuvačić 

Academic Editor

PLOS ONE